# The Implications of Climate Change on Health among Vulnerable Populations in South Africa: A Systematic Review

**DOI:** 10.3390/ijerph20043425

**Published:** 2023-02-15

**Authors:** Myo Myo Khine, Uma Langkulsen

**Affiliations:** Faculty of Public Health, Thammasat University, Pathum Thani, Bangkok 12120, Thailand

**Keywords:** climate change, health impacts, adaptation, poverty, inequality, vulnerable populations, South Africa

## Abstract

Climate change poses numerous threats to human life, including physical and mental health, the environment, housing, food security, and economic growth. People who already experience multidimensional poverty with the disparity in social, political, economic, historical, and environmental contexts are more vulnerable to these impacts. The study aims to identify the role of climate change in increasing multidimensional inequalities among vulnerable populations and analyze the strengths and limitations of South Africa’s National Climate Change Adaptation Strategy. A systematic review was applied, and literature from Google, Google Scholar, and PubMed, as well as relevant gray literature from 2014–2022 were reviewed. Out of 854 identified sources, 24 were included in the review. Climate change has exacerbated multidimensional inequalities among vulnerable populations in South Africa. Though the National Climate Change Adaptation Strategy has paid attention to health issues and the needs of vulnerable groups, the adaptation measures appear to focus less on mental and occupational health. Climate change may play a significant role in increasing multidimensional inequalities and exacerbating health consequences among vulnerable populations. For an inclusive and sustainable reduction in inequalities and vulnerabilities to the impact of climate change, community-based health and social services should be enhanced among vulnerable populations.

## 1. Introduction

Poverty is multifaceted, encompassing more than the inadequacy of wealth, income, and capital to sustain long-term living. Food insecurity, lack of access to quality education and other necessities, lack of social support and prejudice, and not being involved in decision making are all examples of poverty. In 2017, 9.2% of the global population, those who made less than US $1.9 per day, lived under the international poverty threshold [1]. Additionally, the COVID-19 pandemic has threatened long-term poverty reduction and triggered an economic crisis with extreme consequences. Without sufficient responses, the combined effects of the pandemic, its economic crisis and climate change would have a detrimental effect on humans and increase long-term economic costs [1]. Prior to the pandemic, roughly 10% of the world’s population was living below the poverty line, unable to meet even the necessities including access to healthcare, water and sanitation, and education [2].

Climate change, which potentially exposes humans to pandemics and epidemics, is intrinsically linked to poverty. Scientific evidence indicates that the global average temperature is rapidly increasing as a result of excessive greenhouse gas emissions. Moreover, the effects of climate change are being felt globally and are being exacerbated by severe and extreme weather events that occur frequently. Climate change is impacting human life in a variety of ways, and those living in poverty are disproportionately being affected Approximately 79% of the global poor reside in rural areas and rely on climate-sensitive environmental assets such as lakes, forests, and oceans for their livelihood [1]. Globally, 51.6 million people are struggling to cope with the effects of floods, droughts, and storms while also attempting to control the COVID-19 pandemic and cope with its impacts [3]. Anthropogenic climate change is having an impact on 80% of the world’s landmasses, under which 85% of the people live. In comparison to high-income countries, low-income countries have substantially more distributed impacts [4].

By ratifying the Paris Agreement, many countries have set their own targets, specifically, Nationally Determined Contributions (NDCs) for greenhouse gas (GHG) mitigation. They have also devised strategies and policies aimed at mitigating the effects of climate change and the disparities that result from it, along with improving health and socioeconomic outcomes. Nevertheless, there are still major gaps in fulfilling the climate targets, and inequalities persist as a result of the Paris Agreement’s inability to make the targets mandatory, as well as countries’ policies, failing to consider the interconnectedness of various dimensions that influence the social determinants of individual’s health outcome [5].

South Africa is amongst the top levels of extreme poverty and unequal countries in the world, which makes the population of the country sensitive to the effects of climate change. The combined effects of political, geographic, and social factors rank it 92 out of 181 countries in terms of its vulnerability to the effects of climate change [6]. Moreover, the country is experiencing significant climate change impacts particularly caused by higher temperatures and decrease rainfall patterns. The existing vulnerabilities to natural disasters such as strong storms, flooding, and droughts are likely to worsen. Environmental changes in South Africa create adverse effects on multiple factors, air quality, heat, weather events, food security and the burden of diseases. In 2015, 4% of deaths were attributed to air pollution. Furthermore, food security is jeopardized, with crop yields expected to fall across the country, accompanied by livestock failure [7], exposing about 25% of the country’s population to food insecurity [6]. In addition, climate change has a significant impact on water resources, which increases disproportionate health risks for rural communities and vulnerable populations including the poor, women, children, marginalized ethnic groups, migrants and internally displaced individuals [8]. In South Africa, approximately 300,000 people per year are anticipated to be vulnerable to malaria by 2050, under a high-emissions scenario [9].

Climate change response strategies are becoming critical worldwide. In South Africa, adaptations to the effects of climate change also support comprehensive actions to reduce poverty and inequality. The country has consolidated climate change response strategies with this development model. Water resources, land management, agriculture and health are critical strategic areas of focus for the sustainability of the environment [9]. South Africa’s actions and responses could provide gaps and key important points that other countries can learn from, based on their geographical location, specific impact sectors and the level of climate risk.

Climate change is a global health issue and related social determinants such as health, education and food security are being adversely affected, with the poor bearing the brunt of the combined effects. As a result, if countries do not have resilient and inclusive strategies in place to adapt and reduce the effects of climate change, many interconnected dimensions of climate change including health, poverty, hunger and gender equality will be jeopardized, further widening the inequality gap among population groups in the world and the long-term effects for vulnerable populations will be immense. Thus, perpetuating a vicious cycle of poverty with reduced coping abilities. Along with global trends of climate change, South Africa is a unique setting for studying climate change-related issues and analyzing national government adaptation strategies as well as the consequences, as the country has one of the highest levels of socioeconomic and health inequalities in the world.

Therefore, assessing the role of climate change and how it contributes to multidimensional inequalities among populations and identifying the strengths and limitations of national climate change response is critical to identify the gaps and relevance of the responses, in addition to the health implications of climate change on vulnerable populations. The result of this study will be useful in supporting government agencies, environmental activists, and researchers in developing inclusive and resilient climate change adaptation strategies that will enhance the health and social well-being of vulnerable populations.

### 1.1. Climate Change and Poverty

Climate change trends are varied in different parts of the world due to their geographical locations and uneven production of greenhouse gas emissions. However, the effects are distributed evenly around the globe, regardless of where the emissions originate. Additionally, the vulnerability of climate change is influenced by poverty which is increasingly recognized as a dynamic and multidimensional condition that is shaped by historical circumstances and by the interplay of individual and community characteristics and larger social, economic, political, and environmental factors [10]. Changes in climate variability and trends are a growing concern that are likely to exacerbate the existing high vulnerability of improvised households, worsening the prevalence, intensity and persistence of poverty in developing countries. Poorer individuals with low-income informal or hourly jobs with little protection against climate-related employment disruptions suffer from climate impacts and they are more likely to live in areas with higher exposure to climate extremes. Therefore, a lack of financial and technological means to cope with rising climate risks and disadvantaged people with fewer resources suffer the most from the climate impacts [6].

On the other hand, climate change jeopardizes the long-term goal of eliminating poverty, as poor people and countries are highly exposed to all the climate-related effects, including environmental hazards that destroy resources and livelihoods, agricultural production and triggered environmentally induced health problems [11]. Therefore, climate change poses a serious threat to the world’s poor, with the possibility of pushing over 100 million people back into poverty during the next 15 years, making it more difficult for them to escape from poverty. Furthermore, the underlying economic costs of climate change effects, which are inherently unpredictable, have the potential to endanger development in many countries. Although affluent countries are the main contributors to global warming, developing countries encounter more challenges in dealing with its consequences and their ability to respond to it [11].

### 1.2. Multidimensional Poverty and Vulnerable Communities in the South African Population

The poverty level in south Africa is measured by the South African National Standard. In 2021, a person living in South Africa with less than 890 South African Rands per month (approximately US $62.8) was considered poor. Furthermore, people with a monthly food budget of 624 Rands about US $44 were living below the poverty line. In South Africa, 55.5% of the national population, or around 30.3 million individuals are impoverished, living in the upper level of the national poverty line, whereas 25% of the population, approximately 13.8 million individuals were vulnerable to food insecurity in 2014. People in South Africa are less equipped to deal with the effects of climate change due to the country’s higher poverty rate, and the severe economic effects of climate change have increased, which is likely to exacerbate existing inequality in South Africa [6]. Since poverty is relative and multidimensional, people who live in monetary poverty and have no access to non-monetary resources suffer the most from the effects of climate change.

In this research, the global multidimensional index was used to assess the multidimensional poverty of the South African Population. Multidimensional poverty considers multiple dimensions of well-being and assesses individuals’ intersecting deprivation in three dimensions and ten indicators. These dimensions include health, education and standard of living. The health dimension is based on two indicators: nutrition and child mortality, the education dimension is based on two indicators: years of schooling and school attendance, and the standard of living dimension is based on six indicators: cooking fuel, sanitation, drinking water, electricity, housing and assets. These indicators assess household well-being across different dimensions. Each indicator is weighted equally, with a weight of 1/3. As a result, a household is classified as multidimensionally poor if the deprivation score is 1/3 or higher [12].

According to the most recent publicly available survey data for South Africa’s Multidimensional Poverty Index estimation for 2016, 6.3% (3679 thousand people in 2020) of South Africa’s population is multidimensionally poor, while another 12.2% (7155 thousand people in 2020) is vulnerable to multidimensional poverty. In addition, 39.8% of people are severely impoverished and living in multidimensional poverty, while the data indicates that 18.7% of the population living in monetary poverty [12].

Vulnerable groups are those who are at higher risk of poverty and social exclusion than the general population. They include children, those aged 17 and under; youth—people aged 15 to 34 years; women—the female population; older people—people aged 60 and above; and persons with disabilities—those experiencing various degrees of difficulty in functional domains including seeing, hearing, walking, remembering, concentrating, self-care, communicating and social interaction. Moreover, the vulnerability differs in terms of household characteristics, income, health, education, poverty, economic activity, housing and basic services, and social grants. Additionally, the vulnerability is varied in terms of geographical factors, race and gender [13].

In South Africa, disparities in access to healthcare, resources, income, morbidity and mortality persist in vulnerable populations, particularly along economic, racial and gender lines [14]. Thus, climate change’s direct and indirect impacts aggravate existing inequalities among vulnerable populations by interacting with social factors such as poverty, poor housing, living in low-lying areas, limited access to healthcare and inadequate education, placing them in economically, environmentally, and socially vulnerable positions, making them less capable of anticipating, coping with, resisting or recovering from hazardous impacts. Consequently, the impact of climate change on poverty often extends to the impact of climate change on inequality. The extent to which populations are affected by disasters is not solely determined by their proximity to the disaster’s source but rather by a range of contextual factors and combined impacts of social, economic, political and environmental situations. For example, the well-being of many socially vulnerable populations, particularly rural communities reliant on natural resources, may be jeopardized by a moderate hazard event, and they will primarily face social, economic and environmental challenges [8].

## 2. Materials and Methods

### 2.1. Study Design

Using the Preferred Reporting Items for Systematic Reviews and Meta-Analyses (PRISMA) [15] principles, this study conducted a review of the role of climate change in contributing to multidimensional inequalities and the strengths and limitations of South Africa’s National Climate Change Adaptation Strategy.

### 2.2. Study Area

South Africa is situated at the southern tip of the African continent, stretching about 3000 km from the Mozambican border in the east to the Namibian border in the west, with a total land area of 1,219,602 km^2^ [6]. The Indian and Atlantic Oceans collide at Cape Point, the country’s southernmost point. There are four different regions in the country: “the interior plateau, the eastern plateau slopes, the Cape Fold belt, and the western plateau slopes”, according to the great Escarpment, which is made up of Roggeveld Scarp in the southwest, 1500 m above sea level and the KwaZulu-Natal Drakensberg in the east, which is 3482 m above sea level. Within 100 m of the coastline, a quarter of the coast has been developed, where natural protection of storm surges and rising sea levels have been endangered. Therefore, both people- and private-owned property are susceptible to severe storms and the effects of climate change [6].

As of the year 2020, there are 59.3 million people living in South Africa, make up of four racial groups. According to 2021 South Africa demographic data, Black Africans make up 80.9% of the population in South Africa, followed by Colored people (8.8%), white people (7.8%) and Indian/Asian people (2.5%). By the years 2030 and 2050, that number is predicted to rise to 66.4 million and 72.8 million, respectively. It is an upper-middle-income country with the national Gross Domestic Product (GDP) is US $301.9 billion in 2020, with the average growth of 0.2% in 2019 and 7.0% in 2020. Climate change has primarily hampered South Africa’s overall economic growth, and is projected to continue to hamper economic development, power generation, employment opportunities and increase disparities in the future [6].

### 2.3. Data Selection

#### 2.3.1. Data Sources

Electronic databases: Google, Google scholars and PubMed were used to search for relevant literature and sources such as peer-reviewed articles, original articles and grey literature, especially technical and project reports from government and non-profits organizations, non-governmental organizations (NGOs) and international non-governmental organizations (INGOs) such as United Nations International Children’s Emergency Fund (UNICEF), World Health Organization (WHO), and World Bank from 2014 to 2022.

#### 2.3.2. Search Strategy

The key terms, as well as the Boolean search operators “OR,” and “AND,” are applied to obtain online information and documents for this study. The keywords, particularly “Climate change AND health impacts AND South Africa”, “Climate change AND poverty AND South Africa”, “Climate change AND adaptation in South Africa”, “Climate change AND Multidimensional inequality in South Africa”, “Climate change and Women AND Black people”, “Climate change and rural population AND South Africa,” and “Climate change AND Vulnerable populations AND South Africa” are used interchangeably.

#### 2.3.3. Selection Criteria

The relevant articles were chosen using the inclusion and exclusion criteria listed in Table 1. Studies deemed irrelevant by titles and abstracts were excluded. The remaining studies were then independently reviewed and disputed studies were included or excluded based on consensus.

The quality of the evidence was assessed following the Grading of Recommendations, Assessment, Development and Evaluation (GRADE) methodology [16]. The following criteria were taken into account to grade the evidence: study limitations (i.e., risk of bias), consistency of effect, imprecision, indirectness and publication bias. The quality of each key outcome of the included publications was then rated as “high,” “moderate,” “low,” or “very low.” Each study’s quality was evaluated by the authors, who worked independently. A discussion was used to settle any disagreements that emerged.

## 3. Results

### 3.1. Summary of Eligible Records

Through the literature search of the sources, relevant literature and materials that answered the research question were selected. Data and information were organized using EndNote. All articles eligible for inclusion are from 2014 to 2022. In accordance with the standardized criteria, a total of 854 articles were identified and 34 duplicate articles were removed. After screening 820 by abstracts and titles, 796 articles that are not related to the criteria were excluded and 24 articles were included for the review. The relevant outcomes as reported by the included publications that were relevant to the study were extracted and grouped thematically to suit the objective of the review. The themes were descriptively synthesised based on prior discussion and agreement by the reviewers. The following PRISMA flowchart (Appendix A), Figure 1 describes the screening process of literature for the review.

### 3.2. The Role of Climate Change in Increasing Multidimensional Inequality

#### 3.2.1. The Intersectionality as a Tool in Multidimensional Inequality Analysis

An intersectional approach can be applied in different ways when addressing social inequalities and problems. Intersectionality primarily focuses on a variety of multi-level inter-relationships among social locations, forces, factors and power structures that shape and influence human life. Clearly, this approach can be used when assessing different types of inequalities exacerbated by climate change [17]. Needless to say, existing social inequality caused by poverty is a driving force behind rising multidimensional inequality and climate change inequality. Climate change’s impact on health cannot be separated from the social, economic, cultural, and historical factors that influence health and well-being, and it cannot be explained without taking into account the interconnectedness of such factors and the ways in which they are linked [18]. For example, the impact of climate change can be assessed using socioeconomic status as well as a variety of other inequalities such as gender, race, geographic location and capabilities. Although climate change affects everyone, they differ disproportionately depending on the prevailing inequalities and vulnerabilities in relation to race, class, sexual orientation and gender [19].

According to Islam and Wrinkel’s climate change inequality framework, the relationship between climate change and multidimensional inequality is characterised by three major channels that aggravate inequality among disadvantaged populations: (1) increase in the exposure of the disadvantaged groups to the adverse effects of climate change; (2) increase in their susceptibility to damage caused by climate change; and (3) decrease in their ability to cope with and recover from the damage suffered due the climate change [17]. Evidence shows that vulnerable groups’ exposure to climate hazards is related to where they live. Furthermore, inequality in society forces the vulnerable population to live in areas more vulnerable to climate hazards, increasing their exposure to those hazards (i.e., flooding). At the same level of exposure, susceptibility varies across disadvantaged groups according to living conditions and economic activity. The assets held by the poor (i.e., livelihood, housing stocks, livestock) are short, less resistant, and less diversified. Poor rural households do not own much land and may suffer from lower productivity. A lack of savings makes them more susceptible to changes in food and beverage market prices and temporary disruption of public services such as energy, water provision and public transportation. Finally, considering the same level of susceptibility, disadvantaged populations have less ability to cope with and recover from the damage due to a lack of protection mechanisms such as insurance, property rights and access to common property [17]. When climate hazards strike, disadvantaged groups suffer disproportionate losses of income and assets, including physical, financial, human, and social assets. Therefore, climate change impact can worsen inequality, resulting in a vicious circle of multidimensional inequality among the disadvantaged population [17].

#### 3.2.2. Unequal Impact of Climate Change on Vulnerable Populations

It is commonly recognized that those who are poor and marginalized are severely affected by climate change, and social divisions and power imbalances exacerbate climate inequalities. South Africa presents a unique setting when investigating social and health inequalities [20]. It is a country where centuries of colonial discrimination, followed by the apartheid political system, reinforced inequalities across political, socioeconomic and cultural determinants. Although the apartheid regime was abolished in 1994, these disadvantaged groups, such as Black Africans, including Colored, Asians and mixed-race people continue to face systemic inequalities from years of oppression [14]. This pre-existing social determination influences population health inequalities and poses a challenge to South Africa’s Climate Response Strategy. South Africa’s vulnerability to climate change is significantly increased by economic inequality, poverty and reliance on coal-fired power generation, all of which exacerbate existing inequalities [21]. Economic inequality in South Africa is closely connected to the dimensions of class, race and gender. On average, female workers receive 30% less in wages than their male counterparts and male employees are more likely to be hired and have higher-paying jobs. About 75% of the population in the urban poor communities and rural areas are still living below the poverty line with a higher unemployment rate. Furthermore, racialized inequality exists in the labor market, with black people earning lower pay when employed, whereas white people earn significantly higher pay rates than all other population groups. These kinds of disparities make it difficult for poor people to escape from the interconnected dimension of poverty, which in turn marginalizes them from equal access to power and resources, which directly influences the social determinants of health, making them more vulnerable to climate change and perpetuating inequalities. Therefore, understanding inequalities in human wealth, power, and social advantage is critical for understanding climate disruption and how to address the potentially devastating consequences of climate change [20].

##### Gender Impact and Vulnerability: Destructions Affected by Climate Change

Gender intersects with climate risks and vulnerabilities, and gender continues to be a crucial component in defining social structures and identities. Women in South Africa suffer the most from the effects of the climate crisis due to their economic marginalization, political exclusion, and distinct labor responsibilities [19]. In terms of gender dimension, 60–80% of women in South Africa are involved in the agricultural sector for their livelihood and their families depend on their supply and the farm’s productivity, both of which are highly vulnerable to negative impacts of climate change [22]. This condition, when combined with climate change and other social determinants, contributes to a decreased financial status, reduced capacity to provide income for their children and families, increased climate-related gender-based violence (GBV) and food insecurity [19]. Male-headed households own larger plots of land and have access to agricultural credits compared to female-headed households, which limits the poor women’s participation in agricultural activities and places them at an additional disadvantage, such as engaging in multiple part-time activities to make additional income for their households [23]. This unequal access to land among women in South Africa makes them structurally disadvantaged in terms of agricultural production. Conjoined with this inequality, climate change in South Africa has caused crop failures and a decline in agricultural production. It has also resulted in a decrease in on-farm employment, leaving women farmers with a loss of income in addition to rising food prices [24]. Approximately 53.1% of female-headed households experience chronic food insecurity and income uncertainty, while 44.1% of male-headed households are food insecure, with limited access to adequate food and thus experiencing hunger [23].

Additionally, all these economic and social stressors resulting from climate change have forced female workers from the agricultural industry to relocate to urban regions or abroad in search for alternative sources of income. As a result, there is an increase in Gender-Based Violence (GBV) and human trafficking in the country, which puts women at greater health risk. A previous study revealed that climate-related migration creates risks of experiencing human trafficking and sexual exploitation among women. South Africans account for approximately 62% of trafficking victims, with most of them being women, who are commonly exploited in sex and forced labor while also frequently subjected to violence [24].

Moreover, food insecurity caused by climate change is a mediated factor that forces migration among women in South Africa [25]. It has been observed that climate-related female migrants in South Africa have relatively few alternative livelihood options, apart from sex work, due to a lack of education and lack of jobs in the country. They are more likely to engage in sexual work, which poses an increased rate of HIV transmission among women. They often experience GBV [7]. Women in informal settlement areas, on the other hand, have higher HIV prevalence due to a lack of access to HIV prevention and treatment services, and the damage to transportation and health infrastructure due to climate extreme events. The risk of HIV transmission has increased in the country, in which 8 million people are infected with HIV, the highest number in the world with women accounting for 14.4% (60% of all cases) compared to 9.9% of men in 2014 [24]. The geographical distribution of HIV reveals a higher prevalence in rural areas, particularly among black poorer women who live in informal settlement areas. This demonstrates the connection of race and gender, whereby black women who are socioeconomically disadvantaged bear the brunt of the diseases due to complex social and historical inequalities and factors such as stigma, poverty, and poor health have already weakened their resilience to climate change [24].

A case study indicated that women carry additional burdens, such as traveling further to collect water for agricultural and daily use during droughts in South Africa. This kind of additional burden also contributes to GBV among girls and women in the country. The rate of GBV and crime against women in South Africa have increased from 0.9%, accelerated to (8.2%) and (9.6%) in 2015 and 2016, respectively. The country has the highest rate of GBV in the world, affecting one in every three women during their lifetime. This in turn increases the mental illness among women caused by climate change and many other factors, including poverty, inequality, HIV, and the political turmoil of the country [7].

##### Race and Class Inequalities and Vulnerability

The level of vulnerability often varies in terms of race and class, and they are closely interrelated with the dimension of poverty [17]. In South Africa, apartheid created social inequalities such as racial segregation, working-class exploitation, rising poverty, unemployment, and wealth inequality, as well as environmental racism with the unequal enforcement of environmental laws, to name a few. Since then, both poor and black people have been forcibly relocated to climate-risk areas, such as coastal regions and urban industrial communities with poor air quality, where they are exposed to high levels of hazardous pollution. These consequences of social inequalities remain among the different racial groups and classes [26]. In the contemporary context of South Africa, combined with these legacies and its ongoing inequitable distribution of natural resources and power imbalance contributing to environmental racialization, where both black people and people of color continue to live in the most climate-risky areas. In addition, they are placed to work in the climate destruction process such as mines, coal-fired power stations, steel mills, incinerators, and waste sites or polluting industries, where they are paid low salaries. They lack access to clean air and water, electricity, and sanitation [27]. A case study in Gauteng province showed that 1.6 million black African people are residing in mining dumps in the Gauteng province that are contaminated with uranium and hazardous heavy metals such as arsenic, aluminum, manganese and mercury [26]. Likewise, another study showed that the production of toxic gases and dust in the air in mining dumps has increased the prevalence of respiratory diseases among lower-income populations who were exposed to mine dumps. The prevalence of asthma (17.3%), chronic bronchitis (13.4%), chronic cough (26.6%), emphysema (5.6%), pneumonia (17.1%), and wheeze (24.7%) in the exposed communities in 2012, with the higher prevalence among elderly women [28].

Access to domestic supply networks and essential services for black populations is challenging whereas white people can have full access to those services. Consequently, for example, a lack of access to water services forces people to use unsafe water for daily use such as cooking and drinking, posing a serious threat of epidemics and other waterborne diseases. Additionally, a lack of other social determinants such as access to inadequate healthcare, emergency services and adequate schooling makes these population groups more vulnerable to the climate crisis [29].

Currently in South Africa, communities with lower socioeconomic status are the least likely to have green space, trees and infrastructure, and most of the informal settlements are inhabited by non-white residents. Studies showed that white communities with a six-fold increase in income have 11.7% more tree coverage and 8.9% higher vegetation than Black, Indian and Colored communities [14]. This inequality in green spaces contributes adversely to climate-related risks such as air pollution, climate exposure and heat waves. While these green spaces serve as important social and cultural factors, in South Africa, they also serve as a source of income for poorer communities through agricultural use [14].

Nowadays, climate change in South Africa is intensifying due to the changing political situation, thus the working-class people and informal settlements are experiencing devasting impacts including food insecurity due to crop failure and higher food prices, heat-related problems with higher energy prices, water shortages and displacement [25]. A study explained that rising temperatures have become a significant issue in workplaces. Outdoor workers suffered a variety of heat-related impacts, including sunburn, insomnia, irritation and weariness, making it difficult to maintain work capacity and production in extremely hot weather [30]. Likewise, people of color and disadvantaged vulnerable groups are still experiencing the ongoing and intersecting impacts of climate change since they experienced group-based discrimination and are excluded from a variety of contexts. Eventually, they are systematically and structurally excluded from participating in political discussion, healthcare, personal security and education, which in turn increases the multidimensional inequalities and limits their capabilities of adaptation in society [25].

### 3.3. The Overview of Climate Change Adaptation Governance in South Africa

South Africa’s national climate change government has evolved over the last two decades, as stated in the National Development Plan 2030, with a complex process and details of management policies, strategies, regulations and institutions. The national climate change government was established and launched in 2004, with the development of the National Climate Change Response Strategy. Following that, South Africa’s National Climate Change Response White Paper (NCCRWP) was developed and approved in 2011, providing a foundation for the country’s national climate policy. South Africa’s climate governance appears to have made significant progress since 2012, when climate change adaptation and mitigation issues were consolidated as major issues in the National Development Plan 2030 [31].The South African Department of Environmental Affairs (DEA) serves as a coordination body in the development of objectives, frameworks, and implementation of the national climate change policy [32]. Although the DEA oversees the development of national climate change policy, the goals of the National Climate Change Response Policy were guided by other national and international commitments, such as “the South African Constitution, the Bill of Rights, the National Environmental Management Act, the Millennium Declaration and commitments made under UNFCCC” [32]. Multisectoral collaboration of the climate policy is designed, to ensure that all sectors and levels of government are involved in policy development. NCCRWP recognizes the importance of policy alignment from national to local and across national departments in promoting a low-carbon society that is sustainable, just, and adaptable. The South African government established the Department of Cooperative Governance and Traditional Affairs (CoGTA) to support cooperative governance at the national and sub-national levels, it also monitors municipal integration and cooperation, although municipalities are responsible for addressing certain adaptation challenges [32]. South African governance appears to practice a bottom-up approach, with the three levels of government (national, provincial and local) having the authority to impose their own rules and regulations for climate change adaptation while cooperating and collaborating to work together. This approach is different to the policy during the apartheid era, when levels of government were considered to operate hierarchically, and national policy was not commonly implemented at the provincial and local levels. This was a hierarchical approach to governance resulted in policy incoherence, with frequent policy divergence between different governmental levels. The bottom-up governance approach is more democratic since it allows local governments to respond to their own contextual issues with less influence from structural bureaucracy. This approach also makes it easier to collect data and information of the climate-affected individuals, regarding their needs and contexts [33]. Moreover, this enables local governments to adapt their implementation considering the contextual circumstances and to prioritize beneficiaries’ involvement in decision making, which will further help identify the most vulnerable populations and manage the adaptation needs in a way that is just fair and equitable for them [33]. NCCRWP mentions that the bottom-up approach is important for climate adaptation and invites for a coordinated approach from the three levels of government; however, implementations of those policies in their local context are not mandatory [34]. Municipalities are responsible to provide accountable government, whereas the provincial governments are responsible for ensuring that municipal governments follow their regulations and standards, and only intervene when serious concerns about climate governance arise [35]. Furthermore, intervention and actions are advised to be taken by implementation measures, adaptation, and mitigation approaches after carefully considering the “special needs and circumstances” of the most vulnerable populations and fulfilling the best solutions by collaborating with possible actors, institutions, and sectors [31].

#### 3.3.1. Analysis of the National Climate Change Adaptation Strategy (NCCAS) Strengths on Mitigating Health Impacts of Vulnerable Populations

The National Climate Change Adaptation Strategy framework focuses on four strategic objectives, nine intervention points and twelve outcomes for climate resilience development. The guiding principles reflect the vision of the NCCAS and the effort to reduce the vulnerability of climate change, as defined in the key strategic intervention priorities and expected outcomes. The strategic points are outlined to respond to and protect the climate change impacts in a cohesive manner [36]. NCCAS recognizes the country’s structures and dimensions of various remaining inequalities which resulted from the policies of apartheid. NCCAS acknowledged that historical injustices and social inequalities are the foundations of vulnerabilities; thus, climate adaptation solutions must take into account the situations and factors that make people more vulnerable to the impacts of climate change [37]. By focusing on key adaptation sectors such as water, health, human settlements, agriculture and commercial forestry, biodiversity and ecosystems, and disaster risk reduction and management, NCCAS addresses the social, structural and political factors that contribute to inequity and inequality. It also highlights the importance of considering gender, age, wealth, social status and other factors in the measurement of adaptation strategy in order to take equitable adaptation actions [36]. Among the 12 guiding principles of the NCCAS, the principles consider vulnerable populations in the development of the strategies which includes consideration of vulnerable group, gender-responsive, based on the best available science and traditional knowledge, and equity. In terms of principles, “Consideration of vulnerable group” takes into account how climate-related disasters affect people’s health, particularly the health of those who are most vulnerable, such as women, children, the impoverished people who live in climate-prone and rural areas, and sick and disabled people. Gender-responsive focuses on expanding women’s involvement in climate change decision making, measuring various climate change vulnerabilities, addressing needs and prioritizing both men and women to reduce gender inequalities [36]. The guiding principles also mention that the implementation of NCCAS will take into account both scientific and traditional knowledge for the response. In terms of equity, the strategy aims to promote equity while providing environmental protection [36]. After analyzing each intervention point and key strategic outcomes, the strategy focuses to mitigate the health implications of climate change on vulnerable populations. NCCAS seeks to address health implications through the lens of socioeconomic and sociopolitical aspects of vulnerability by focusing on investing and strengthening multiple sectors. Particularly, the framework focuses on promoting the livelihoods of rural people, women and vulnerable households through capacity-building activities such as climate-smart and conservation-farming techniques, water-saving practices and building climate-resilient structures, while also attempting to reduce carbon emissions through these activities. Along with these activities, it seeks to restore natural resources and seeks to address gender differences in terms of access to and control over natural resources. Therefore, these kinds of responses could create the development of policies which promote gender equality and climate-resilient livelihood opportunities for the population, which contribute to reducing many health challenges caused by socioeconomic inequality among vulnerable populations [36]. However, the actual intervention of these actions has not yet been recognized.

The framework aims to identify the underlying causes of vulnerability and gender inequality, and their related risks and implications to individuals within communities and municipalities. By addressing these issues, the framework focuses on designing and implementing vulnerability reduction programs, as well as increasing the capacity of emergency departments, human resources, and healthcare infrastructures to respond to climate-related incidents [36]. The objective of the framework is to focus on improving disease monitoring and surveillance systems, and developing early-warning systems for climate disasters and their projected effects on agriculture, disease-transmission patterns, and air-quality information. The Infectious Diseases Early Warning System project was developed and launched in the country to a develop strong early-warning system for all climate-related infectious diseases, particularly to forecast changes in the malaria incidence and outbreak expansion in different areas of the country [30]. As demonstrated in a study conducted in Cape Town, careful monitoring of the air temperature can forecast peaks in the incidence of diarrhea disease, enabling health facilities to plan for increases in hospital admissions and outpatient visits [38]. Moreover, NCCAS proposes climate services to be developed in key vulnerable sectors with the provincial and local municipal government participation, and implemented in areas where risks have been identified, as well as among vulnerable population groups such as rural farmers and coastal communities [36]. Evidence from earlier collaborations demonstrates a decrease in the prevalence of infectious diseases such as malaria, diarrhea and pneumonia, as well as their associated mortality rates. However, no statistical data supporting this claim has been presented [37]. The framework also highlights that women are more susceptible to the health effects of climate change because their poverty rate is higher than that of men, compounded by climate-related disasters. Women are more severely affected by the indirect health effects of structural and systemic discrimination, unequal access to resources as a result of power disparities, and unjust societal roles and responsibilities than males are. To protect women from hunger and malnutrition, the framework seeks to expand and promote food gardening programs in the community and household food gardens, in areas that are not classified as agricultural land. South Africa has also begun social interventions aimed at women, such as offering relief kits and house-building grants. Moreover, the principle also calls for women to be presented in the process of decision-making in agricultural improvement [30].

#### 3.3.2. Limitations of the National Climate Change Adaptation Strategy in Practice

South Africa is making significant progress, at both research and applied levels with the understanding of the climate change impacts, and the adaptation strategy has been well framed and approved to implement and make responses [36]. However, there are some limitations of the strategy to actually implement into practice: (1) gaps related to the insufficient assessment of the impact and data information gaps on the socioeconomic implications related to climate change, and (2) institutional gaps and challenges to collaboratively work with the public and private sectors for the adaptation priorities.

##### Gaps Related to the Insufficient Assessment of the Impacts and Data Information Gaps

Although NCCAS has been modelled and released since 2017, many gaps and limitations still remain. Firstly, the adaptation priorities are difficult to implement, because it appears that the assessment of socioeconomic vulnerabilities and adaptation responses are developed without having comprehensive knowledge of the forecasting chain of climate projections and their associated impacts. The vulnerable populations and the apartheid legacies of social inequalities are repeatedly addressed with the inclusion of vulnerable people, women, children and disabled people; however, black people and other disadvantaged racial minority groups are not emphasized [36]. Moreover, the effectiveness of adaptation strategies among women and vulnerable population groups in the country is not mentioned, and strategic actions to adjust adaptation accordingly are not addressed [30].

While the importance of women’s participation and gender equality is addressed in the majority of the adaptation priorities, the specific mechanism on how they will be implemented is not mentioned. Of greater concern, the intervention actions listed all the important stakeholder’s participation in decision-making, but vulnerable population participation is not specifically stated in the framework [36].

South Africa lacks a strong database system for extensive climate data, which makes the framework difficult to obtain national climate change and generate through the development framework. There is a shortage of data and information regarding the connections between impacts and climate change [36]. Although the country has the South African Risk and Vulnerability Atlas (SARVA) system which the health risk, driver, exposures, and vulnerabilities can be analyzed, the lack of a strong scientific climate database and health data system hindered to prepare specific actions on forthcoming climate health impacts [30].

Furthermore, the strategy itself notes that it is difficult to estimate the cost of adaptation under the climate change scenario due to the uncertainty of climate variability and a lack of data about the impacts [36]. The strategic actions lack knowledge and inadequate research regarding how extreme weather occurrences, coupled with informal settlement growth, could drive up infrastructure costs in South Africa [30]. A lack of data about climate change impacts on vulnerable populations creates inaccuracy in estimating the adaption cost and in identifying feasible finance solutions, and creates a significant barrier to undertake specific implementation. A lack of credible comprehensive data on the current climate change creates challenges for adaptation and undermines the legitimacy of policy [32].

##### Institutional Gaps and Challenges of the National Climate Change Adaptation Strategy

The cross-sectoral implementation plans are well demonstrated in the framework; however, some institutional challenges and gaps remain in terms of climate governance structures; communication issues between different tires of government, the capacity of the human resources, and public–private engagement issues.

*Complex government structure of climate change*: South Africa technically has a well-developed government structure in place that outlines the policies to be consistent between municipalities and at all levels of government, from the national to the local. Many multisectoral intervention actions and intervention points are proposed for implementation by a multilevel government from various sectors in the NCCAS framework. A strong collaboration and coordination between them may be challenging when it comes to bringing these actions into practice [36]. The policy formulation and implementation are difficult due to the complex nature of the climate responsibility of governments, which includes the poor integration of adaptation policies across jurisdictional levels, poor involvement of stakeholders and poor integration knowledge. Poor integration among different governments often results in a lack of clarity regarding the policies, which makes it difficult to agree on how different policies will be jointly implemented [32].

*Limited human and financial resources and public sector capacity*: A study in eThekwini Municipality, KwaZulu-Natal province mentions that there are very few civil servants who work as climate change advocates and at the local government departments. Poor black women found it difficult to report drought problems and to share their experiences on climate impact issues [33]. South Africa has very limited expertise in tackling climate-related issues, and key several departments are understaffed [39]. The government’s ability to deal with climate change and related policies is limited by a lack of human and financial resources, and a scarcity of appropriate expertise and assistance [32]. A lack of sufficient human resources in South Africa’s health system has hampered efforts to prepare the health system resilient to extreme weather events and forthcoming climate-related infectious illnesses. A lack of human resources frequently creates a considerable limitation in gathering data on climate change-related health impacts, which makes it relatively difficult to perform a health impact assessment on vulnerable populations [30].

A lack of financial resources often leads to poor financial aid provision to research institutions, researchers and practitioners. Of greater concern, there are fewer research outputs related to health and climate change approximately only 3% of the publications mention health impacts out of 600 publications in 2015 [30]. When developing and implementing sector, cross-sector transactions to resilience policies, and the lack of human and financial resources become even worse at municipal and provisional levels. Thus, sufficient human resources, financial resources and experties with sufficient skills are challenges for policy generation in South Africa [32].

*Weak public, private sectors, and communities’ engagement*: The framework calls for multisectoral approach to reduce the vulnerability of climate change on vulnerable populations. Adaption actions are well proposed with a set of time frames and call for public-private sectors to be implemented collaboratively. It also mentions that the participation of stakeholders from public and private sectors, including communities, government, NGOs, Community-based Organizations (CBOs), Civil Society Organizations (CSOs), and business entities are important in carrying out systematic interventions [36]. Many stakeholders in South Africa, including the government, CSOs, researchers, practitioners and the private sector, have poor relationships when it comes to generating climate change solutions. These relationships arise as a result of different perspectives on vulnerability, timing, policy structure, concern about how they would handle information and feedback from various stakeholders [39]. The private sector in South Africa, in particular, finds it difficult to collaborate with other stakeholders and to adopt climate-resilient adaptation actions in their operations, and they are likely to require incentives to do so [30].

Additionally, these weak relationships resulted in a lack of integrated and consistent information, an insufficient assessment of the impacts of climate change scenarios, socioeconomic impacts, and vulnerability. Therefore, cross-sector integration adaptation is difficult to implement, and the prospective impacts of individuals and sectors will not be clearly identified, making it difficult to implement the strategic actions for health impacts within the given time frame [32]. On the other hand, the inconsistencies of assessment create uncertainties and difficult to identify the impacts on vulnerable populations [38]. For example, the framework addressed the different vulnerabilities of the different levels of settlement; however, it did not mention any certain actions or a separate plan for providing infrastructure and services to rural and urban settlements in accordance with their various needs due to insufficient assessment of the vulnerability [33].

## 4. Discussion

### 4.1. Multidimensional Inequality in the Lens of Intersectionality

Understanding intersectionality entails realizing that women’s vulnerability is mostly a result of the societal duties and responsibilities connected with their identities as well as the “power imbalances that deny them equitable access to resources, knowledge and education”. Given the socioeconomic disparity of apartheid in South Africa, the vulnerability and multidimensional inequality of poor women and impoverished racial groups are determined by their existing socioeconomic factors such as poverty, low income, discrimination, inadequate and poor housing infrastructure, and degraded locations intensified their vulnerability to climate change. When all these factors were combined, they experienced increased multidimensional inequality, which may not be felt by those who live in cities with access to essential services. Several complex systems, for example, health inequalities, discrimination and racism are multidimensional processes that impose cumulative disadvantages and advantages through access to power structures [18]. This rationale is relevant to the fact that the power dynamic of the black racial group and poor economic class individuals, that they were subjected to environmental racism during apartheid laws, as well as structural discrimination in terms of the settlement, access to primary services and employment. As a result, those who are wealthier will probably have more resources to adapt to climate disasters, while most of the poor and those living in climate risk areas are vulnerable to climate change and will suffer from combination effects due to their social and historical background, lack of material resources and vulnerability. Additionally, there is a strong correlation between race and gender, that racial discrimination makes black women more disadvantaged in terms of productive resources and power to ensure to adapt the impact of climate change and greater the existing vulnerabilities. Articulating vulnerability through the intersectional lens will help in identifying how climate change can exacerbate the existing multidimensional inequality of vulnerable populations.

### 4.2. Strengths in Mitigating the Health Impacts

The effectiveness of adaptation is dependent on lowering poverty and inequality, particularly among women and other vulnerable populations. The implications for vulnerable people: immigrants, women, children and other racial groups, must be considered when developing strategies for extreme events and certainly for all climate interventions [30]. This is relevant to the NCCAS guiding principles of reducing poverty and promoting equity through empowering women and other vulnerable populations. South Africa has a poverty reduction program intended for susceptible populations, which is relatively helping to improve their living standard and reduce health risks. Furthermore, there is a need to do impact assessments and investigate them through the lens of racial inequality to determine how such impacts potentially harm marginalized racial groups. Early warning and monitoring, disease surveillance, and the capacity building of the health system and healthcare workers have been effective, and some infectious diseases have been reduced at a national scale. However, some part of the country still suffers from climate-related health impacts. A study suggests that focused research and the effective use of surveillance data are required to monitor climate change’s impacts and the traditional strengths of the country’s health sectors [30]. Since this effectiveness and appropriateness of implementation are already recognized, these interventions can be expanded at all areas of the country especially in climate-prone areas to get disaggregated health impacts data on vulnerable populations. A strong monitoring of undernutrition, along with agricultural productivity is important regarding the susceptibility of South Africa’s food security to climate change.

The complicated relationship between health and climate change should be understood, substantial health risks indicated by hospital admission and mortality should be documented, and then health impacts can be presented using this evidence [39]. However, the framework did not mention detailed health risks, for example, heat stress for agricultural workers and informal settlement households. The documentation of the relationship between health risks and heat stress is in the ideal stage of the strategy. Thus, climate-related health risks should be well documented by improving disease surveillance data systems to reduce the health impacts among vulnerable populations.

Economic empowerment and building resilient farming techniques for vulnerable populations are also one of the strong points of the strategy as the health outcomes are determined by the economic situation of the people. By promoting the livelihood of vulnerable populations, the combined social stressor caused by socioeconomic inequality might reduce among them. On the other hand, recognizing the different vulnerabilities of males and females is another strength that the framework illustrated. Regarding the recognition, the evidence of offering housebuilding grants is likely to be effective. The framework seems to be uncertain about implementing food gardening initiatives for women to prevent malnutrition and hunger.

### 4.3. Limitations Related to Insufficient Impacts Assessment and Data Information Gaps

The NCCAS does not specify vulnerable population participation in the decision-making of the adaptation strategy, but it rather mentions that vulnerable populations are the core of adaptation. Such generalizations will limit the participation of the most vulnerable populations in decision-making. Therefore, their adaptation, knowledge and experience of vulnerability may be overlooked when developing the solutions [30]. The comprehensive knowledge of the forecasting chain of climate projections and their associated impacts, the apartheid legacies of social inequalities and socioeconomic vulnerability should be well assessed in developing climate adaptation strategy. A case study of UK climate change management shows that UK Climate Change Impact Program (UKCIP) is a dedicated support body to assist with impacts and adaptation assessments. UKCIP provides a guidance framework to the stakeholders on how to approach impacts and adaptation assessments, to pilot innovative approaches to adaptation, and to review and reflect on best practices. This program consistency was discovered in a few settings, including the UK Climate Change Risk Assessment 2017 and a flood risk management plan for London, in terms of climate change risk management with multicriteria analysis [40]. This kind of mechanism and support body might be relevant for South Africa’s NCCAS to accurately perform impact assessments and adaptation assessments, and to develop an appropriate strategy accordingly. Meanwhile, adaptation uncertainties can occur if the country does not have a strong database system. Credible data is essential to engage with local government about the relevant problems and created the best solution [41]. In South Africa, a lack of this robust database system effect generating climate information and assessing the impacts, which occur uncertainties in adaptation actions and further hinders effective implementation. This makes it difficult for decision makers to identify sufficient resources for adaptation.

### 4.4. Limitations Related to Institutional Challenges

In the adaptation policy, collaboration efforts of the different levels of government are essentially important in terms of knowledge sharing and policy integration. A lack of cooperation between them even leads to information and knowledge gaps and the adaptation efforts often overlooked the needs of the local community [42]. However, South Africa’s complex government system lacks coordination between tires of government and creates challenges to bring the climate change issue into collective management. This lack of cooperation between government levels may make it difficult to adopt a strategy that encompasses health and other disciplines. A shortage of financial resources limits the provision of funds to research organizations; hence, high-quality and evidence-based data information might be frequently constrained for a management plan. Sufficient financial and human resources are very important for the government to implement sectoral and multisectoral adaptation actions to reduce health impacts in South Africa. This also requires strong climate data and high-quality evidence research on climate change and health [30]. South Africa needs a comprehensive strategy for climate finance and a plan outlining how resources would be distributed for implementing climate change solutions and attracting foreign investment [32]. Similarly, it is very important to have sufficient human resources in research institutions, government departments, and health systems to manage disease surveillance and collect climate-related health data.

Moreover, findings show that South Africa public–private relation regarding climate change is very weak and a lack of cooperating sufficient assessment on the impacts and vulnerability. A strong relationship between all the stakeholders and strong vulnerability assessment are key important things of adaptation strategy. However, most of the vulnerability assessments are often limited by the involvement of diverse groups of stakeholders and are rarely followed by adaptation interventions. Building a shared understanding of climate risk and social inequity between officials, practitioners, academics, and the vulnerable themselves can help to capture an understanding of vulnerability that can then be easily integrated into adaptation action plans [41]. Thus, having a climate change government system is not enough to do quality and multisectoral impact assessment for policy making, strong communication between different stakeholders, including between information producers and the user is critical and relevant knowledge from diverse stakeholders should be integrated. Participation of the vulnerable communities can provide qualitative stories and quantitative information, which researchers can integrate and share them between multiple stakeholders from different sectors for quality assessment.

### 4.5. Limitation of the Study

Many researchers have identified a variety of ways in which climate change may contribute to vulnerabilities; however, the majority of empirical research particularly at the aggregate level in South Africa has concentrated on food and agriculture, the limited literature with disaggregated data on the health impacts of climate change could potentially be a limitation. Therefore, it is necessary to conduct further studies on the impacts of climate change on economic growth and poverty traps and its possible resultant vulnerabilities to health. Additionally, the study could be limited by the inclusion of only publications in the English language. Since the study is dependent on the quality of the original data collected, this could possibly limit its quality as well. The review may have missed out unpublished work as it included only publications available in the databases used.

### 4.6. Recommendation

Strengthen early-warning systems for climate stressors, identify climate-sensitive conditions early, and plan for the early mobilization of disaster and emergency response services, as well as continuous evaluation for effective adaptation.Governments should establish a policy to construct a comprehensive national health monitoring and surveillance program that monitors and evaluates indicators related to the health consequences of climate change at all levels.Governments should create a coordination finance mechanism in the NCCAS’s priority actions, which help coordinate current financing sources and generate new sources. In this way, the central government could also assist the provincial and municipal levels by increasing their financial capacities, providing guidance for project preparation and the allocation of targeted funding.Governments should increase investment in both action-oriented adaptation research and impact research, as well as collaborative partnerships with international researchers to generate evidence and develop capacity in climate change research.Healthcare systems and infrastructure should be strengthened to be resilient and more accessible to the vulnerable population by supporting community-based health and social service organizations to enhance community awareness of climate risks among vulnerable populations.

## 5. Conclusions

This study analyzes South Africa’s Climate Change Adaptation Strategy, along with the role of climate change in increasing multidimensional inequality through gender and race dimensions, and how that impacts human health. The links between climate change and poverty and vulnerability are complex, multidimensional, and context specific. Climate change has enormous impacts on human life and health, and those impacts are disproportionately bigger among women and black racial groups with lower economic status and many existing social inequalities. Social, economic, political and health inequalities are closely linked to climate change, they need to be recognized as a part of the problem and included in solutions. A holistic approach will consider the effects of climate change already on different populations, how poverty, gender and race inequality left by apartheid influenced the climate change impacts, and what to learn from past developments and the current situations. This would lead to a transition towards a more sustainable and safer future for everyone. Sustainable Development Goals (SDGs) constitute a solid basis to align climate action, as they portray many areas which are and will increasingly be affected by climate change, such as health, poverty, hunger, migration, water, sanitation, and gender and race equality. Therefore, it is essential to not regard these dimensions as separated from each other, but to develop an understanding of their interconnectedness and the complex relations between them. To address one of them means addressing all of them. This requires adequate finance plan, transparency and collaboration in the assessment, implementation, monitoring and evaluation of solutions by all actors.

## Figures and Tables

**Figure 1 ijerph-20-03425-f001:**
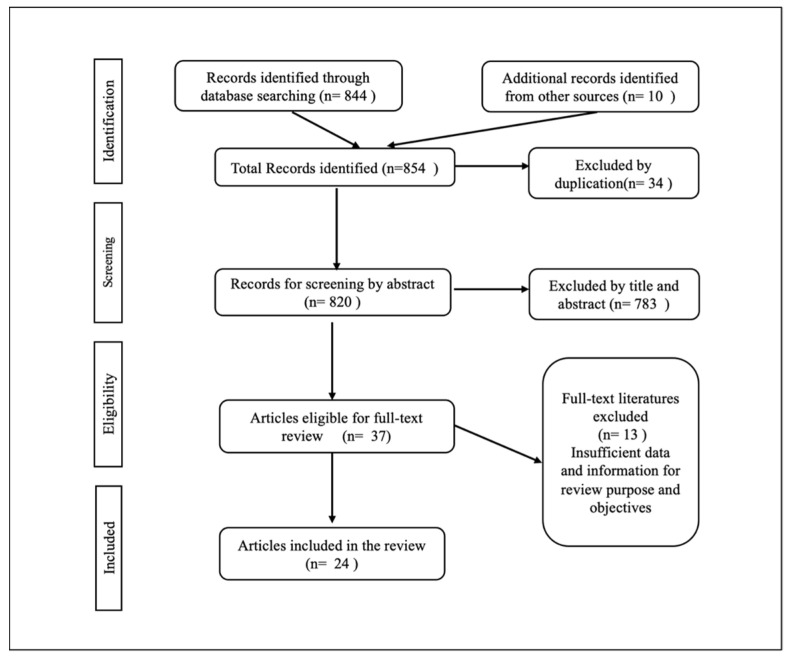
Study selection.

**Table 1 ijerph-20-03425-t001:** Inclusion and exclusion criteria of the study.

Inclusion Criteria	Exclusion Criteria
Peer-reviewed articles published between 2014 and 2022 that contain relevant information relating to the review’s purpose and objectives.	Articles that discuss health implications of climate change on the general population without a specific focus on the vulnerable populations.
Peer-reviewed articles and gray literature with relevant data and information about the study question and objectives in the abstract and full text are available.	Articles that only describe the climate change and response without any analysis of the potential health implications and inequalities.
Peer-reviewed articles, gray literature, and technical reports that provide either qualitative or quantitative data information on South Africa’s climate change response and health implications on vulnerable populations as well as climate change and its associated dimensions	
Publications on vulnerable populations such as women, disadvantaged racial and ethnic groups, and people of lower socioeconomic status were considered for inclusion.	
Literature which is only written in English.	

## Data Availability

Not applicable.

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
