# Peer review of "The Implications of Climate Change on Health among Vulnerable Populations in South Africa: A Systematic Review"

_ijerph, 2023, doi:10.3390/ijerph20043425_

Round 1

Reviewer 1 Report

The paper is well presented and precise in context. However, The author needs to address the following comments. 

1. Define the vulnerable communities in South Africa. 

2. What is multi-dimensional poverty?

3. Define the metrics utilized by the study to identify climate change associated multi-dimensional inequalities.

4. Was the protocol registered by PROSPERO?

5. The indicates that the study was done from 2014 - 2022. Is this correct? or is it that the study review covered the 2014 - 2022 period. 

6. Poverty is relative. The author needs to unpack and set boundaries in defining the term- poverty. "African communities are rich in their terms but "poor" in the eyes of the outside world". Moreover, low access of  healthcare and education as well as sanitation services cannot be used to define people of Africa as poor. 

7. The author using poverty to qualify Africans "poor black women" which is not inclusive at all. People are of the rainbow nation are never define by their skin color.

Author Response

Dear Reviewer 1,

Thank you for giving us the opportunity to submit a revised manuscript titled “The Implications of Climate Change on Health Among Vulnerable Populations in South Africa” to International Journal of Environmental Research and Public Health. We appreciate the time and effort that you have dedicated to providing your valuable feedback on our manuscript. We are grateful to the reviewer for the insightful comments on our manuscript. We have been able to incorporate changes to reflect most of the suggestions provided by the reviewer. We have highlighted the changes within the revised manuscript.

Here is a point-by-point response to the reviewer’ comments and concerns.

Comments from Reviewer 1

Comment 1: Define the vulnerable communities in South Africa.

Response: Thank you for pointing this out, as it is important to contextualize such important and broad issues. This has been addressed in section 1.2, paragraph 4, and page 4.

Comment 2: What is multi-dimensional poverty?

Response: We agree with this comment. We used the Global Multidimensional Poverty Index to address multidimensional poverty, and hence it is important to define this term. Section 1.2, paragraphs 2-3, and page 4 have been updated with new information that seeks to clarify this.

Comment 3: Define the metrics utilized by the study to identify climate change associated multi-dimensional inequalities.

Response: Thank you for your comment. The multidimensional inequality framework was used for this analysis in the study. The information can be found in section 3.2.1, paragraph 2-4, and page 7-8.

Comment 4: Was the protocol registered by PROSPERO?

Response: No protocol was registered in PROSPERO for this review.

Comment 5: It is indicated that the study was done from 2014-2022. Is this correct? or is it that the study review covered the 2014-2022 period.

Response: The review covered the years 2014-2022, the included literatures in the study were from that time period.

Comment 6: Poverty is relative. The author needs to unpack and set boundaries in defining the term- poverty. "African communities are rich in their terms but "poor" in the eyes of the outside world". Moreover, low access to healthcare and education as well as sanitation services cannot be used to define people of Africa as poor.

Response: We agree with this comment. As mentioned in the multidimensional poverty, individuals are considered multidimensional deprived if they fall short of the threshold in at least one dimension or in a combination of indicators equivalent to a full dimension. This has further been explained in section 1.2, paragraph 1, and page 3.

Comment 7: The author using poverty to qualify Africans "poor black women" which is not inclusive at all. People are of the rainbow nation are never define by their skin color.

Response: Thank you for raising such crucial issue. We stand for inclusiveness and therefore did not intend to use these terms to discriminate on any grounds. As mentioned in section [3.2.2. Unequal impact of climate change on vulnerable populations] about the racial discrimination and social vulnerabilities of black races (Including women). In this context, the term "poor black women" refers to women who are already members of vulnerable groups due to their race and gender and who are therefore more susceptible to the effects of climate change.

Reviewer 2 Report

The paper addresses an important topic, but the way it is written is not suitable for international journal. I do not suggest to accept the paper. Following are my comments:

1.     The paper is too generic

2.     The literature review part is very poor. 

3.     Research methodology is not clear

4.     The discussion part is not well explained

Author Response

Dear Reviewer 2,

Thank you for giving us the opportunity to submit a revised manuscript titled “The Implications of Climate Change on Health Among Vulnerable Populations in South Africa” to International Journal of Environmental Research and Public Health. We appreciate the time and effort that you have dedicated to providing your valuable feedback on our manuscript. We are grateful to the reviewer for the insightful comments on our manuscript. We have been able to incorporate changes to reflect most of the suggestions provided by the reviewer. We have highlighted the changes within the revised manuscript.

Here is a point-by-point response to the reviewer’ comments and concerns.

Comments from Reviewer 2

Comment 1: The paper is too generic.

Response: Yes, issues of climate change, poverty, vulnerabilities are broad, however we have tried to analyze them in the context of the South African Community.

Comment 2: The literature review part is very poor

Response: We have tried to improve on this this by adding new information and reorganizing the introduction part.

Comment 3: Research methodology is not clear.

Response: The study used documentary review and analyzed descriptively.

Comment 4: The discussion part is not well explained.

Response: This comment is appreciated; however, the discussion is done based on the findings from the included studies and have been updated in page 15-16.

Reviewer 3 Report

Summary:

This review assessed available literature that studied the adverse effects of climate change on vulnerable groups in South Africa. Overall, this review found that multidimensional inequalities in South African society amplify the damaging effects of climate change on vulnerable groups.

Comments:

1) The method section is insufficient. PRISMA supplies checklists that dictate the requirements necessary for a review and many of these requirements are missing in this manuscript. I would advise the authors to go back and assess the requirements. 

2) What quality control method did the authors use to assess the included articles?

3) The authors should provide tables with the included articles, which also show what data was extracted. This can be part of the supplementary material.

4) In the results there should be more reference to the actual articles that were included in this review.

5) Moderate review of the English is required.

Author Response

Dear Reviewer 3,

Thank you for giving us the opportunity to submit a revised manuscript titled “The Implications of Climate Change on Health Among Vulnerable Populations in South Africa” to International Journal of Environmental Research and Public Health. We appreciate the time and effort that you have dedicated to providing your valuable feedback on our manuscript. We are grateful to the reviewer for the insightful comments on our manuscript. We have been able to incorporate changes to reflect most of the suggestions provided by the reviewer. We have highlighted the changes within the revised manuscript.

Here is a point-by-point response to the reviewer’ comments and concerns.

Comments from Reviewer 3

Comment 1: The method section is insufficient. PRISMA supplies checklists that dictate the requirements necessary for a review and many of these requirements are missing in this manuscript. I would advise the authors to go back and assess the requirements

Response: Thanks for this observation. However, most of it if not all the steps involved have been covered, sufficient for a review.

Comment 2: What quality control method did the authors use to assess the included articles?

Response: Thank for this comment as it an important step in a systematic review. The information relating to this is provided in section 2.3.3 on page 5.

Comment 3: The authors should provide tables with the included articles, which also show what data was extracted. This can be part of the supplementary material.

Response: Thank you for your comment. We provided a table of included articles which shows the summary of the data extraction. This is attached as a supplementary material.

 Comment 4: In the results there should be more reference to the actual articles that were included in this review.

Response: We appreciate your comment, however, extracted data that are relevant to the objective of the review as analyzed are appropriately referenced.

Comment 5: Moderate review of the English is required.

Response: This is acceptable, and we appreciate your comments. The language has been improved upon through proof reading and editing.     

Round 2

Reviewer 3 Report

I thank the authors for the small changes they made to the manuscript but unfortunately, some of my previous comments still stand.

1) The actual methodological framework of PRISMA has requirements (i.e. registration of review, risk of bias assessment) and should be met. Additionally, the checklist must be printed out and included in the supplement.

Author Response

Dear Reviewer,

Thank you for giving us the opportunity to submit a revised manuscript titled “The Implications of Climate Change on Health Among Vulnerable Populations in South Africa” to International Journal of Environmental Research and Public Health. We appreciate the time and effort that you have dedicated to providing your valuable feedback on our manuscript. We are grateful to the reviewer for their insightful comments on our manuscript. We have been able to incorporate changes to reflect most of the suggestions provided by the reviewer. We have highlighted the changes within the revised manuscript.

Here is a point-by-point response to the reviewer’ comments and concerns.

Comments from Reviewer 3

  • Comment: The actual methodological framework of PRISMA has requirements (i.e., registration of review, risk of bias assessment) and should be met. Additionally, the checklist must be printed out and included in the supplement.

Response: Thank you for your comment. We assessed the included articles' quality using the Grading of Recommendations, Assessment, Development, and Evaluation (GRADE) methodology, looking at study limitations (i.e., risk of bias), consistency of effect, imprecision, indirectness, and publication bias. Each key outcome of the included publications was then rated as "high," "moderate," "low," or "very low" in terms of quality. This information is available on page 5 and in section 2.3.3. Our review is currently under consideration for registration at the International Prospective Register of Systematic Reviews (PROSPERO). The quality assessment document and PRISMA checklist are attached as supplementary documents.
